# Process Optimization of Injection Overmolding Structural Electronics with Regard to Film Distortion

**DOI:** 10.3390/polym14235060

**Published:** 2022-11-22

**Authors:** Martin Hubmann, Behnam Madadnia, Jonas Groten, Martin Pletz, Jan Vanfleteren, Barbara Stadlober, Frederick Bossuyt, Jatinder Kaur, Thomas Lucyshyn

**Affiliations:** 1Polymer Processing, Department of Polymer Engineering and Science, Montanuniversitaet Leoben, 8700 Leoben, Austria; 2Centre for Microsystems Technology, Imec and Ghent University, Technology Park 126, Zwijnaarde, B-9052 Ghent, Belgium; 3Joanneum Research Forschungsgesellschaft mbH, Franz-Pichler Str. 30, 8160 Weiz, Austria; 4Designing Plastics and Composite Materials, Department of Polymer Engineering and Science, Montanuniversitaet Leoben, 8700 Leoben, Austria; 5kdg Opticomp GmbH, Am kdg Campus, Dorf 91, 6652 Elbigenalp, Austria

**Keywords:** injection molding, simulation, in-mold electronics, structural electronic, flexible electronics, overmolding, lamination, multi-layer films, moldflow

## Abstract

The integration of structural electronics in injection-molded parts is a challenging step. The films—comprising of laminated stacks with electronics—are exposed to shear stresses and elevated temperatures by the molten thermoplastic. Hence, molding settings have a significant impact on the successful, damage-free manufacturing of such parts. In this paper, test films with polycarbonate (PC) sheets as outer and two different thermoplastic polyurethanes (TPUs) as middle layers incorporating conductive tracks on a flexible printed circuit board (flexPCB) are manufactured and overmolded with PC. Parameter studies investigating the influence of the melt temperature, mold temperature, injection speed and used TPU layer were performed. The molded parts were inspected visually and compared with a numerical simulation using injection molding software. A shear distortion factor for the TPU layer was derived based on the simulations that linked the shear stresses with the injection time and the softening (melting) of the TPU. The distortion of the films was found to reduce with higher melt temperature, lower mold temperature and faster injection speed. Films using the TPU with the higher melting temperature yielded significantly better results. Moreover, distortion on the films reduced with the increasing distance to the gate and a larger cavity thickness was found to be beneficial. All those relations could be correlated with the shear distortion factor.

## 1. Introduction

In recent years, the integration of electronics in injection-molded parts to manufacture “smart” mechanical, durable and light products cost-effectively has gained traction [1,2]. One way is to adapt the in-mold decoration (IMD) process. Here, a sheet or film is inserted into an injection mold and overmolded by rapidly injecting a thermoplastic melt. As a result, a plastic part with printed graphics and/or high-gloss surfaces is manufactured. By additionally incorporating conductive inks and surface-mounted electronic components to electronic films, the injection molded structural electronics (IMSE) process is formed. The assembly is exposed to high temperatures as well as pressures and shear stresses due to the molten viscous melt during overmolding. For that reason, TactoTek has postulated a certification process for cthe omponents and surface mounting adhesives used in their IMSE designs [3,4,5]. 

To reduce the prevailing forces on the components, Ott and Drummer proposed using thermoplastic foam injection molding. Through this technology, they could encapsulate epoxy-based printed circuit boards (PCBs) with significantly lower cavity pressures (about 1 MPa) compared to traditional injection molding [6].

Structural electronics can also be manufactured by embedding flexible/stretchable PCBs and components via lamination processes between covering plastics sheets [7]. The thus created flat stack might be thermoformed to a 3D-shaped structure before injection overmolding. The stack is then additionally exposed to complex elongations during thermoforming. This can cause the out-of-plane deformation of the laminated structures, such as the conductive tracks. Madadnia et al. [8] were able to reduce this undesirable deflection by adding a fractal structure to the original meander-shaped conductors.

The application and merging of multiple materials to one (functional) object is certainly ambitious from a sustainability point of view. Välimäki et al. [9] recently showed the potentials of incorporating bio-based and recycled-film materials and replacing metals and metal oxides with PEDOT:PSS, carbon and amino acid/heterocycles to save energy and reduce the depletion of limited material resources.

It is difficult to make predictions about the “moldability” of injection-molded parts as multiple physical variables have to be considered. For instance, the viscosity—the resistance towards flow—of most thermoplastics decreases with the increasing shear rate and temperature in a non-linear fashion. Hence, numerical injection molding simulation software has become a must-have tool to make assessments about ever increasingly complex molding tasks in advance.

Kololuoma et al. [10] showed the feasibility of the hybrid integration of printed and flexible electronics in combination with conventional electronic components in a personal activity meter demonstrator. They overmolded their assembled hybrid activity badge foils with thermoplastic polyurethane (TPU) using two sets of molding parameters. The setting with a lower melt and mold temperature and slower injection speed resulted in broken electrochromic (EC) displays of their devices. Injection molding simulations using Moldex3D (CoreTech System Co. Ltd., Taiwan) revealed higher shear stresses and filling pressures for this set of parameters. The authors concluded that in this case the adhesion between the layers of the EC display was exceeded by the vertically acting shear stresses.

Numerous authors have looked at different aspects of IMD-produced parts to optimize the molding parameters to reduce high scrap rates inherent to this sensitive process [11,12,13,14,15,16,17,18]: The insertion of the film on one side of the mold can cause an asymmetric flow front advancement as heat transfer is retarded on that side [12,17]. The insulating effect of the film frequently causes warpage due to an unsymmetrical temperature profile across the part thickness during packing and cooling [15,16]. An asymmetrical cooling system with a lower mold temperature at the side of the film can significantly reduce the warpage [12,13]. Part geometry [18] and film thickness [13,16] as well as processing settings, such as mold and melt temperature [12,16], will determine the final dimensions of the warped parts.

Liu et al. [14] systematically investigated the effect of injection molding processing parameters on the ink wash-off during the IMD process. To that end, they used 20 wt.% glass-fiber-filled polyethylene terephthalate (PET) to overmold cut PET films (250 µm in thickness) in a rectangular mold (80 × 20 mm^2^). The film featured screen-printed grid patterns (2 × 2 mm^2^) and the size of the washed area was evaluated. An experimental design using the Taguchi method was established containing the five parameters of melt temperature, mold temperature, melt injection speed, injection pressure and part thickness (0.5, 1 and 2 mm, respectively). They concluded that wash-off could be reduced by adopting a larger part thickness, a lower mold and melt temperature as well as a lower injection speed.The same tendencies were reported by Woyan et al. [16] who evaluated the washed-out ink on polycarbonate (PC) films under the injection point when overmolded in a rectangular plate mold with PC. The analysis of variance (ANOVA) of their full factorial design however did not yield any of the factors as statistically significant. They further compared two PC grades, with the material with the better flowability demonstrating very low ink wash-off. They concluded that the wash-off mainly depends on the shear stress, and hence a thick part and/or low-viscosity melt would decrease the shear stresses on the ink. Furthermore, the melt solidifies on the film during filling, forming a frozen melt layer that connects with the ink. The layer that is stretched by the melt can cause ink delamination. The higher the melt temperature is set, the thinner the frozen layer becomes and the easier it stretches.

The harsh overmolding conditions can damage the structural electronics, lead to high scrap rates or even make the process unfeasible (cf. Figure 1).

To the best of our knowledge, there are only limited studies in the literature related to the optimized injection molding process for flexible laminated electronics. This paper aims to gain a better understanding of how the molding settings impact the flexible laminated electronics, and hence to develop molding guidelines. This is achieved by combining the observations of performed experiments with numerical simulations.

Test film strips resembling a typical laminated flexible electronics stack were fabricated and overmolded with polycarbonate (PC) in a stepped plate mold. Two different TPU types were used as glue layers and the processing parameters, melt and mold temperature, as well as the injection speed, were investigated using a two-level full factorial design of experiments (DoEs). The produced parts were then visually examined for damage. Injection molding simulations using Autodesk Moldflow Insight 2021 (AMI, Autodesk Inc., USA) were performed to virtually inspect the films regarding the prevailing temperatures and shear stresses during overmolding. Finally, a shear distortion factor (*f_τ_*) was derived based on the simulation results.

## 2. Materials and Methods

### 2.1. Film Design

Laminated stacks, as shown in Figure 2a, were prepared and cut into 115 × 30 mm^2^ film strips (Figure 2b) for overmolding. Two different layer structures were assembled and are shown in Figure 2c,d, respectively. The outer layers consisted in both cases of 125 µm-thick PC sheets (Isosport Verbundbauteile GmbH, Eisenstadt Austria). Here, a 5 × 5 mm^2^ grid was screen-printed on both sides of the melt facing sheet (bottom side) using the non-conductive ink NORIPHAN^®^ HTR N 990/010 NC—Tiefschwarz (Proell GmbH, Weissenburg, Germany). A Upisel SR1220 film (50 μm PI—18 μm RA Cu from UBE EXSYMO CO., LTD, Tokyo, Japan) was used as the middle layer that could serve as a flexible printed circuit. In the first stack, the 100 µm-thick TPU Bemis 3914 (Bemis Associates Inc, Shirley, MA, US) was utilized, and in the second stack, the 75 µm-thick Platilon U073 (Covestro AG, Leverkusen, Germany) was chosen as an adhesive between the layers. The Bemis 3914 and Platilon U073 TPUs in the following will be referred to as TPU B and TPU P, respectively. A Lauffer laminator press (Maschinenfabrik Lauffer GmbH & Co. KG, Horb am Neckar, Germany) was used to laminate the different layers of the stacks with lamination parameters as indicated in Figure 2e. 

Figure 3 shows the specific heat capacity curves (*c_p_*) of the individual film layers measured using a differential scanning calorimeter DSC1 (Mettler-Toledo International Inc., Columbus, OH, US). The semicrystalline TPU P exhibits a more pronounced melting peak at higher temperatures compared to TPU B (enthalpy: 11.2 vs. 0.8 J/g and peak temperature: 163 vs. 123 °C). The melted fraction, *α_m_*, of the melting peaks are also depicted. TPU B is fully melted at around 150 °C, which is ~40 K below the melting peak of TPU P. The amorphous PC layer yields a glass transition temperature T_g_ ≈ 150 °C.

### 2.2. Part Geometry and Molding Material

The film strips were overmolded within a stepped, rectangular, plate mold in which the wall thickness reduced from 3 to 1 mm, as shown in Figure 4. The films were fixed within the mold at the fixed half by applying a temperature-resistant adhesive tape on the flow-front-facing film edge (this tape and the flexPCB are not shown in Figure 4). 

A polycarbonate PC Lexan OQ1028 (Sabic, Riyadh, Saudi Arabia) of high optical quality was used for molding. A total of 0.5 wt% of yellow CC10104356BG masterbatch (PolyOne Color & Additives Germany GmbH, Melle, Germany) was added to color the PC and thus make the interface to the overmolded film more clearly visible.

### 2.3. Injection Molding and Experimental Design

A fully electric Arburg Allrounder 470 A Alldrive (Arburg GmbH + Co KG, Loßburg, Germany) injection molding machine equipped with a 25 mm screw was used to perform the overmolding. A Wittmann Tempro plus D 160 (WITTMANN Technology GmbH, Wien, Austria) temperature-control unit was utilized to regulate the mold temperature. The dosing volume was set to 40 cm^3^ and the switchover point (velocity to pressure-controlled filling) was adapted for each setting. The packing pressure was set to about 80% of the filling pressure for 10 s and the residual cooling time was set to 50 s.

Preliminary tests were performed for both stacks with TPU B and TPU P using different injection speeds ranging from 10 to 70 cm^3^/s at a melt temperature of 340 °C and a mold temperature of 120 °C. Those tests already indicated an important role of the TPU-layer: Films using the TPU B showed damage below injection speeds of about 60 cm^3^/s, while good parts could already be produced at injection speeds of 30 cm^3^/s, if films were used with TPU P in their stack.

For further investigations, the 2-level full factorial designs of experiments (DoEs) were created. In such parameter studies, input variables (factors) are investigated at two set levels (low and high) and all possible factor (*k*) combinations are investigated in a total of 2*^k^* runs.

Individual DoEs for both TPU types with factors of melt temperature (A), mold temperature (B) and injection speed (C) were created and are shown in Table 1. DoE I features films using TPU B in its stack (setting B02-B10) and DoE II films with TPU P (P02-P10). The low-level injection speed (C) of DoE I was chosen to correspond to the high-level injection speed (C) of DoE II. Thus, DoE III (B01-B05 and P01-P05) can be derived with factors of melt temperature (A), mold temperature (B) and TPU type (D).

Additional center-point (CtPt) runs were added in which all the factors of the corresponding DoEs ere set to the intermediate level (CtPt DoE I: B10, CtPt DoE II: P10, and CtPt DoE III: B01 and P01).

Three parts per setting were produced, which resulted in a total of 3 × 2 × (2^3^ + 1 + 1) = 60 parts produced. They were evaluated according to the procedure explained in the following: Section 2.4.

### 2.4. Molded Parts’ Analysis Procedure

It was not possible to “measure” the appearance of the films according to some measured quantity. In such cases, Kleppmann [19] suggests to establish a graded evaluation scheme based on a subjective assessment of the parts. In this way, it should still be possible to perform statistically sound comparisons. 

Hence, the overall appearances of each of the overmolded films were assessed by separating the produced parts into three groups: I—not or slightly damaged, II—clearly damaged and III—severely damaged. In addition, the— counted via visual inspection—fraction of distorted squares (*f_s_*) was determined. Finally, the damage of the parts was weighted by introducing a distortion factor *f_d_* = *f_s_* × *c*. Where *c* corresponds to the assigned damage category: *c_I_* = 1/3, *c_II_* = 2/3 and *c_III_* = 1. This was conducted to capture the distortion of the produced parts in a more realistic manner.

Those (unitless) distortion factors (*f_d_*) were then used as the output variables (responses) to evaluate the three DoEs presented in Table 1. To that end, the statistic software Minitab (Minitab Inc., State College, PA, US) was used. It performs a linear regression, linking the investigated factors of the DoEs (input variables) with the distortion factor (*f_d_*, output variable or response). The response of each factor was hereby modeled linearly. In order to detect possible nonlinearities, additional center-point settings were added as described above. 

The software also estimates which of the model factors are statistically significant based on the concept of the analysis of variance (ANOVA). One output of this analysis is the *p*-value. It is calculated for each factor indicating the risk to reject the null hypothesis (no relationship between factor and response) when in fact the null hypothesis is true. Frequently, a factor is considered as significant (and not of random origin) for *p* ≤ *α* = 5%, with *α* being the significance level [19]. A *p*-value for the center point (CtPt) is also calculated, thereby indicating if at least one of the factors included in the model behaves in a nonlinear manner.

### 2.5. Simulation Model Preparation

The commercial injection molding simulation software Autodesk Moldflow Insight 2021 (AMI) was used to numerically study the molded DoE (cf. Table 1). It provides an insight into the temperatures and shear loads faced by the film during filling. As a consequence, a simulation can help in developing a more profound understanding of the process. 

To model the filling phase, AMI numerically solves the conservation equations of mass, momentum and energy using the finite element method (FEM) [20].

Two 3D FEM models were created featuring either the laminated B-stack or the slightly thinner laminated P-stack film strips. The Upisel layer was excluded from the modeling resulting in a three-layer film of PC–TPU–PC and the injection-molded part, as shown in Figure 4. 

The global edge length was set to 1 mm with a minimum number of 12 elements through the thickness for the injection-molded part (AMI property part). The film gate and the region of the part in contact with the film was modeled with a mesh size of 0.5 mm. A minimum number of 6 elements through the thickness and a mesh size of 0.5 mm were selected for each of the three film layers (AMI property part insert). The auto-sizing scale factor was set to 0.9 and the machine die was modeled as a beam hot runner.

The linear tetrahedral element counts for the B- and P-stack models were 2,766,738/893,954/973,914/924,070 and 2,759,495/896,252/991,487/924,095 (part/PC layer/TPU layer/PC layer), respectively.

The simulation material data (Cross-WLF viscosity coefficients, Tait pvT coefficients) for the PC Lexan OQ1028 overmolding material were provided by Sabic (Autodesk udb-file). The possible influence of the added master batch (e.g., on the viscosity) was not considered.

While the specific heat capacity (*c_p_*) of the individual stack layers was measured (cf. Figure 3), the thermal conductivity (*λ*) and density (*ρ*) of those materials also needed for the thermal analyses were not. Those quantities were approximated by choosing the values given in the AMI material data base for PC Makrolon 2805 and TPU Desmopan 487 (both from Covestro) for the PC and TPU B or TPU P layers, respectively.

A uniform heat transfer coefficient (HTC) of 5000 W/(m^2^∙K) (AMI default for the filling phase) was used for all interfaces.

A starting temperature of 25 °C was assigned to the films and a 10 s contact time with the hotter mold prior to start of injection was specified. This should be about the time it took the operator to start a new injection molding cycle after inserting the film into the mold.

Fill was selected as the analysis sequence and 20 intermediate results were set to portray the process conditions in a higher time resolution. A constant mold wall temperature according to the experimental design was assumed as the surface boundary condition.

### 2.6. Simulation Analysis Procedure

The TPU layer has a significant influence on the overall intactness of the overmolded films (as revealed by the preliminary tests). Distortion will presumably only occur when the TPU becomes molten due to the prevailing temperatures (melted fractions: 0 ≤ *α_m_* ≤ 1). The damage on the film will increase the higher the shear stresses (τ) introduced by the viscous melt become and for the longer they act (*t*). To capture this, a shear distortion factor
(1)$$f_{\tau }=\int_{0}^{t_{fill}}\tau \left(t\right)\;\cdot \;\alpha_{m}\left(t\right)\;dt\;\left(Pa\;\cdot \;s\right),$$
was derived. It is used as a magnitude to assess and compare the shear-introduced loads on the film between the different settings.

A Python script was developed that made use of the Synergy Application Programming Interface (API) [21] to access the AMI results and information about the node location and element allocation: The closest shear stress at wall (τ) results of the part mesh were projected to the TPU-layer nodes. Similarly, the temperature results of the TPU-layer nodes were used to calculate the melted TPU fraction (*α_m_* illustrated in Figure 3) during filling (*t_fill_*). Hence, using Equation (1) and the Python scipy.integrate.quad numerical integration routine [22], an individual shear distortion factor ($\bar{f_{\tau }}$) for each TPU-layer node could be obtained.

The procedure for obtaining *f_τ_* is illustrated in Figure 5. Finally, the local shear distortion factors (*f_τ_*) were drawn as shaded contour plots using the API. An overall averaged shear distortion factor () for the TPU layer was calculated as well.

## 3. Results and Discussion

### 3.1. Characterization of Molded Parts

Good reproducibility (pooled standard deviations s_p, TPU B_ = 0.03 and s_p, TPU P_ = 0.03) was observed between the three parts produced per molding setting (as mentioned in Section 2.3 above). Hence, only one overmolded part is depicted per molding setting in Table 2 and Table 3 for films using TPU B or TPU P, respectively (settings corresponding to Table 1). The averaged distortion factors *f_d_* and categorizations (as defined in Section 2.3) are stated in Table 2 and Table 3, too.

The produced parts that are outlined in Section 2.3 representing different DoEs with three repeats each were analyzed using a statistic software (Minitab). ANOVAs were performed with the distortion factors *f_d_* chosen as the response. Only the (stronger) main effects were investigated and the interactions were neglected in the linear regression. This was performed to not overinterpret the subjectively obtained distortion factors *f_d_*. The coefficients of determination of the regressions were at least R^2^ ≥ 80%. The results of the ANOVAs are summarized in Table 4. 

The *p*-values of the coefficients (A, B and C) for DoEs I and II given in Table 4 are below the frequently used significance level of *α* = 5%, indicating their statistically significant influence. Additionally, the *p*-values of the center points (CtPts) for DoEs I and II are small, suggesting that at least one factor behaves in a nonlinear fashion.

For DoE III given in Table 4, only the coefficients for mold temperature (B) and TPU type (D) were found significant. This cannot be concluded for the melt temperature (A) and the center point (CtPt) at a significance level of *α* = 5%.

Figure 6 shows the factorial plots that display the relationships between the response and individual variables. In each plot, the calculated response is plotted when the investigated factor is set to its low and high levels, while the other factors are set to the intermediate level (blue lines). In addition, the center-point setting is drawn in red, which would correspond with the average response (gray, dashed line) for a complete linear case.

Figure 6a (DoE I) and b (DoE II) show that the films are less distorted (lower *f_d_*) the higher the melt temperature, the lower the mold temperature and the faster the injection speed. The *f_d_* is lower for a film using TPU P compared to one with TPU B when overmolded using the same settings, as can be seen when observing factor D in Figure 6c (DoE III). 

The most critical section of the overmolded film is the area where the mold thickness reduces to 1 mm. Here, the most extreme distortion of the screen-printed squares can be observed (Figure 7b–d). As a general observation, the farther away from the gate, the less distorted the film appears. Areas far from the gate are exposed to hot temperatures and shear stresses induced by the melt for shorter periods of time. The section of the film closest to the gate is within the 3 mm part of the mold and features the least damage. Here, the lower prevailing shear stresses (due to the thicker remaining flow channel) seem to outweigh the fact of longer exposure to the hot melt. Yet, distortions can be observed in this area in some cases as well (Figure 7a).

There seems to be an “in-wash” effect of the screen-printed squares towards the center of the film for areas of the film farther from the gate (Figure 7e). This is likely due to the faster progression of the melt front to the sides of the film where the cavity thickness is not reduced.

More severely damaged films also sustained wrinkles in the flex-board (Figure 7f).

The squares were screen-printed on both sides of the melt-facing PC layer (cf. Section 2.1). Hence, shear deformation within that PC should be detectable when the top and bottom lines of the layer would not align any more after overmolding (Figure 7c). This was, however, not the case for the vast majority of the lines; thus, arguably, the whole PC top layer was affected by deformation. As already established, films using the TPU P in their stack yielded significantly better parts than those molded with the same settings using TPU B. This middle layer that is not in direct contact with the melt apparently influences the integrity of the PC layer that is in contact with the melt. It seems that when the TPU layer heats up during overmolding, it loses stiffness and causes the whole stack to crumble. Seemingly, TPU B is more/sooner affected by this than TPU P. The influence of the thickness differences between the stacks using TPU B and TPU P is addressed in Section 3.2 below.

### 3.2. Analysis of Simulation

Figure 8 shows the simulated shear stresses for the settings with low melt and mold temperature (cf. experimental designs in Table 1). 

The thickness difference of 50 µm between the different stacks appears to only have a minor effect on the shear stresses. The calculated shear stresses in case of the slightly thinner TPU P stack (larger remaining cavity thickness) of setting P02 are ~1% lower than that of B02 using the same process settings. Henceforth, one can neglect the difference in the stack thicknesses when comparing them.

Shaded contour plots of the (local) shear distortion factors (*f_τ_* in Equation (1)) are shown in Table 5 (stacks with TPU B) and Table 6 (stacks with TPU P) for all the tested molding settings (as declared in Table 1). In addition, the averaged shear distortion factors ($\bar{f_{\tau }}$) of the TPU layer are given for each setting.

There seems to be good overall correlation between the observed distortion on the molded parts depicted in Table 2 and Table 3 and the shear distortion factors (*f_τ_*) plotted in Table 5 and Table 6, respectively. For instance, $\bar{f_{\tau }}$ predicts the three settings with the highest distorted films—B04, B05 and B08 for TPU B and P08, P09 and P04 for TPU P—correctly. Additionally, the location of greatest damage within the film was accurately predicted.

From the given results, it is difficult to specify a threshold for the shear distortion factor above which the distortion of the films will be visible. As a rough estimation, parts with shear distortion factors *f_τ,_*_*max*_ ≤ 0.01 Pa∙s appear free from distortion. A comparison of the shear distortion factors between films using TPU B and TPU P is limited. For instance, B04 and P08 feature similar $\bar{f_{\tau }}$, but the former setting using the TPU B in its stack appears more damaged. TPU P might have a higher viscosity once melted compared to the TPU B, and thus might be more resistant against shear induced distortion in the molten state. It should be stressed again that simplifications were made in the simulation models, such as selecting a common thermal conductivity (λ) and density (ρ) for both TPU types and assuming a constant mold temperature.

Table 7 shows the regression equations for the DoE given in Table 1 with respect to the derived averaged shear distortion factors ($\bar{f_{\tau }}$). Interaction terms were added to the regression until a R^2^ ≥ 90% was attained.

The factorial plots depicted in Figure 9a (DoE I) and b (DoE II) show that $\bar{f_{\tau }}$ is influenced in a similar fashion as the visually inspected parts (cf. Figure 6).

The pronounced interaction between the mold temperature and injection speed (BC) is graphically illustrated by the different slopes of the lines in the interaction plot. The physical background might be that the TPU layer needs to be heated up towards its melting temperature first (*α_m_*(T)), which will occur more rapidly when the film is already at higher temperatures due to the higher mold temperature. 

The $\bar{f_{\tau }}$ is lower for a film using TPU P compared to one with TPU B when overmolded using the same settings, as can be seen when observing factor D in Figure 9c (DoE III). The notable interaction between the mold temperature and TPU type (BD) might be explained by the higher melting temperature of TPU P (cf. melting curves presented in Figure 3). 

Nonlinearity was depicted again in all three models, possibly induced by the mold temperature (factor B), which was chosen with a large difference of 60 K between its low and high settings.

### 3.3. Demonstration

We used the proposed approach to optimize a real industrial demonstrator part of the “Smart@Surface” project with a similar stack composition. Figure 10a shows an implementation of a stretchable circuit laminated in between a TPU/PC stack, thus forming a structural electronics device. The circuit includes two regions (circles) to act as touch sensors and contact pads for the assembly of side-emitting LEDs that will be used for the demonstration of light guiding. While the film is severely damaged during overmolding when using setting A (Figure 10b), it is possible to obtain a structurally intact part with setting B (Figure 10c). A summary of the used processing settings is given in Table 8, in which for setting B the abovementioned relations were utilized. Hence, setting A corresponds to the settings of the DoE, which produced bad parts, and setting B corresponds to the settings of the DoE, which produced good parts in order to prove that these settings also produce similar results in the real demonstrator part compared to the simple test part, on which the DoE was performed. However, a high mold temperature was set in both cases to obtain a better surface finish.

## 4. Conclusions

The individual layers of structural electronics had an influence on the feasibility to injection overmold them without defects. It was not only the layer of the film that was in direct contact with the hot melt that was affected by it during filling, but also the layers beneath. Two TPU types with different melting characteristics were used to serve as the middle layers of the investigated composite films, which had a significant influence on the produced parts. A softening (melting) of those internal layers can trigger severe distortions on the films, induced by the shear stresses of the viscous melt. Consequently, damage to the films can be reduced or avoided by choosing appropriate molding parameters. In our case, choosing a low mold and a high melt temperature as well as a high injection speed proved beneficial. In general, regions of the film located within the thicker section of the mold and farther from the gate were found to be less affected. The process was numerically simulated using injection molding simulation software to investigate the physical principles behind those observed relations. A shear distortion factor (*f_τ_* in Equation (1)) was derived to establish a relation between the film’s temperature (softening), the prevailing shear stresses and the injection time. A clear relationship between a low shear distortion factor (*f_τ_*) and intact overmolded films was observed, which could thus be applicable as a minimization criterion when developing process guidelines. In our case, no distortion was visible on parts with distortion factors *f_τ,_*_*max*_ ≤ 0.01 Pa∙s.

## Figures and Tables

**Figure 1 polymers-14-05060-f001:**
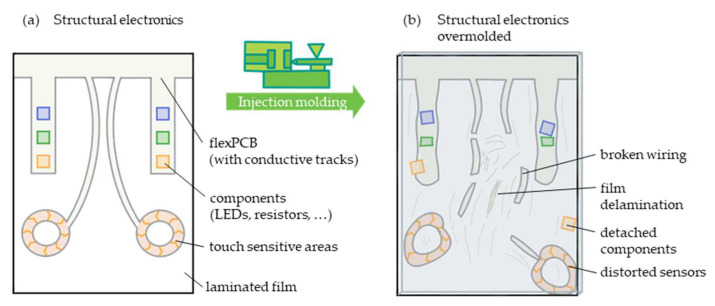
Sketch of a structural electronics part with possible features (**a**) and possible defects after overmolding (**b**).

**Figure 2 polymers-14-05060-f002:**
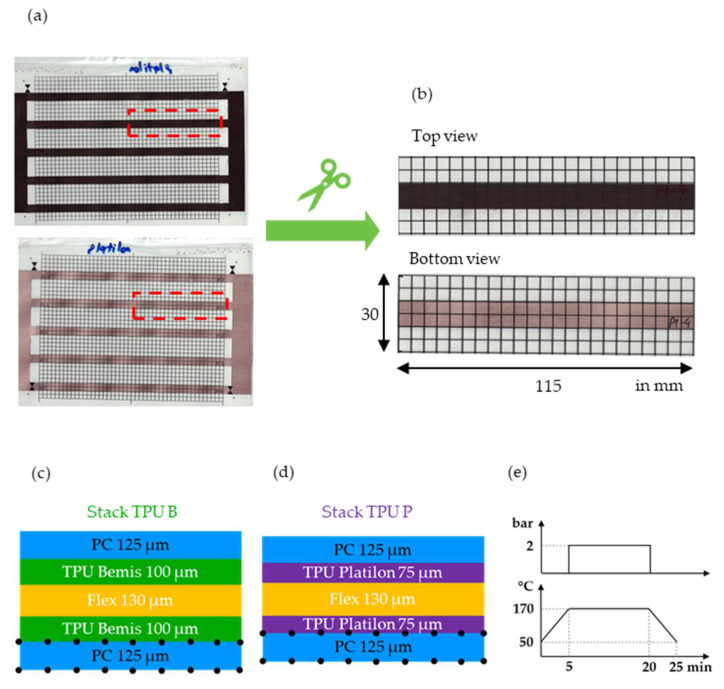
The individual sheets were laminated together (**a**) and cut into the film strips (**b**) for overmolding. The two used layer structures are shown in (**c**,**d**), respectively, and the lamination parameters are indicated by (**e**).

**Figure 3 polymers-14-05060-f003:**
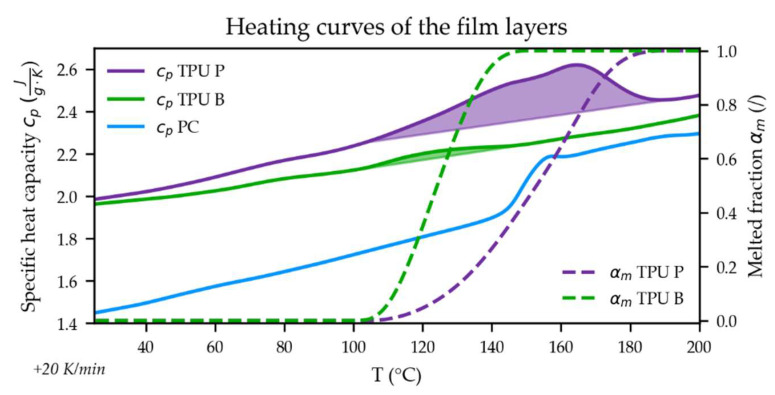
Specific heat capacity curves of the film materials. The melted fractions, *α_m_*, of the semicrystalline TPUs are depicted by the dashed lines.

**Figure 4 polymers-14-05060-f004:**
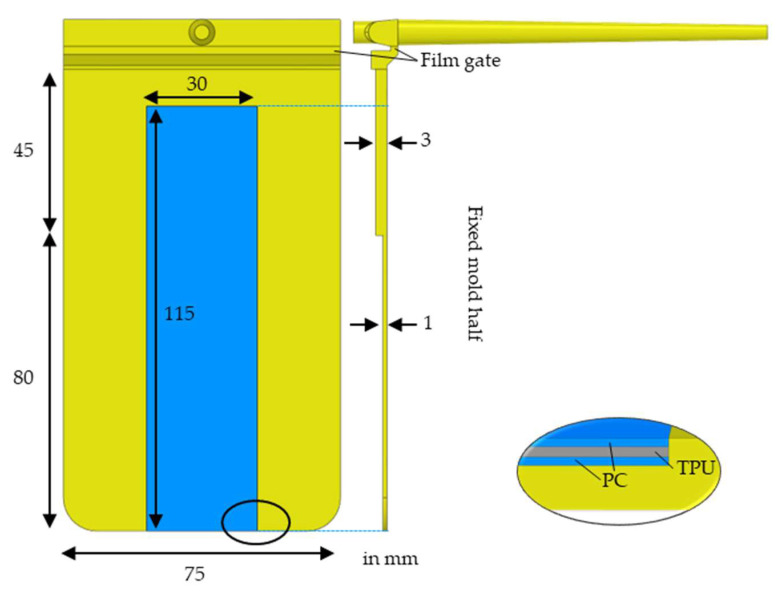
CAD of the stepped plate mold with dimensions. The film is placed on the fixed mold half as illustrated by the blue rectangle. The inset shows the film as modeled in AMI comprising PC and TPU layers (where neither the flexPCB nor the adhesive tape for fixation in the mold are modeled).

**Figure 5 polymers-14-05060-f005:**
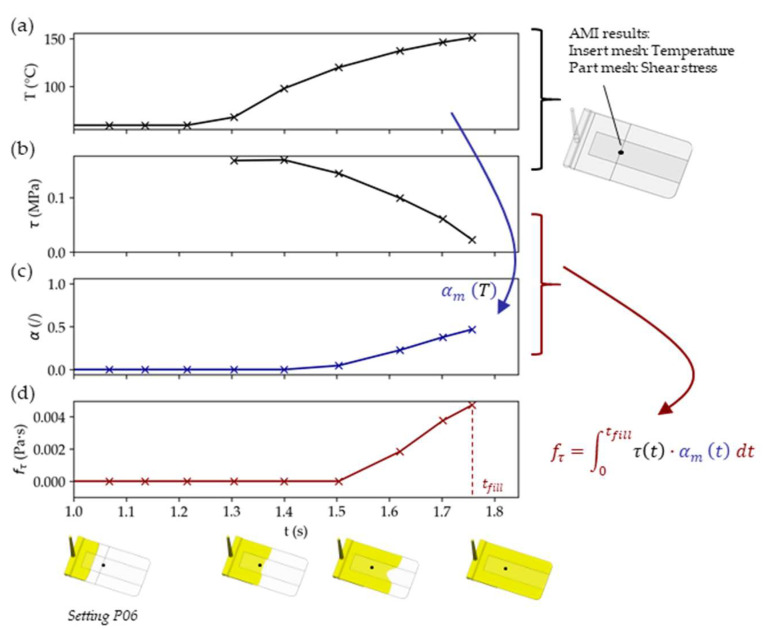
Schematic of the shear distortion factor (*f_τ_*) computation on the TPU layer for a node at the position indicated by the black point: The TPU layer’s temperature (**a**), at first at the set mold temperature, rises once the plastic melt reaches the film, which is also subjected to shear stresses (**b**). The TPU layer softens as indicated by the melted fraction (**c**) and the shear distortion factor, defined in Equation (1), increases until the part is filled (**d**).

**Figure 6 polymers-14-05060-f006:**
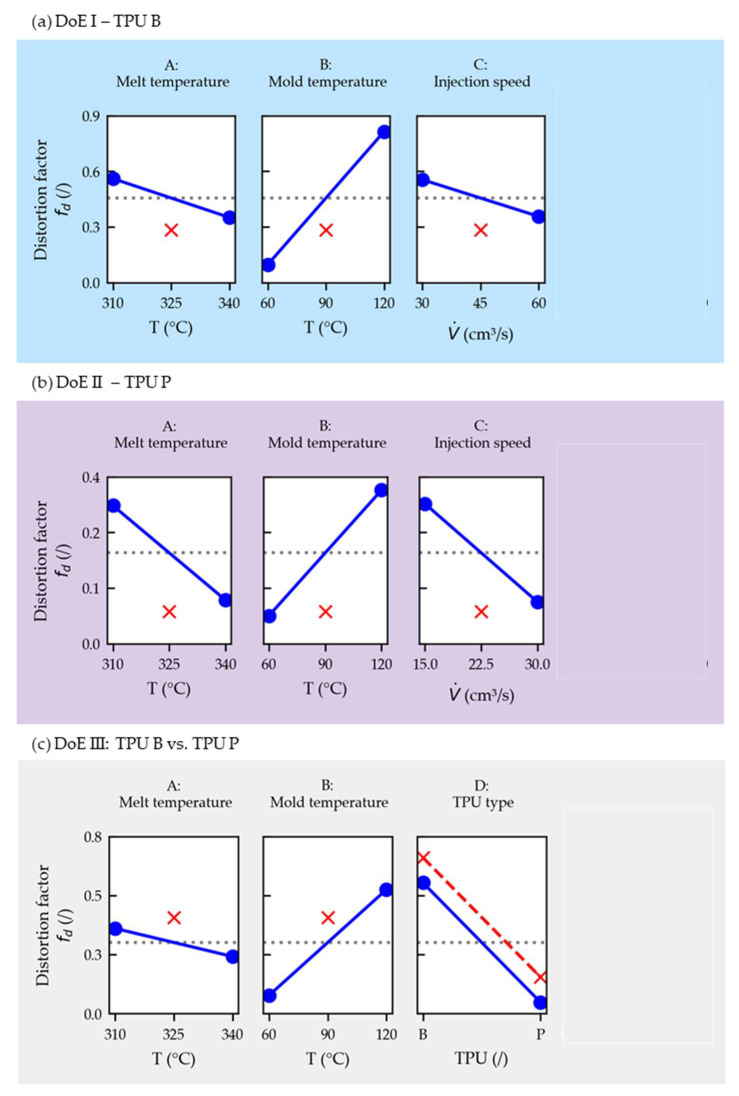
Main effects plots (including center points in red) for the distortion factors (*f_d_*) for DoE I (**a**), II (**b**) and III (**c**). The factor designations (A, B, C and D) relate to Table 4.

**Figure 7 polymers-14-05060-f007:**
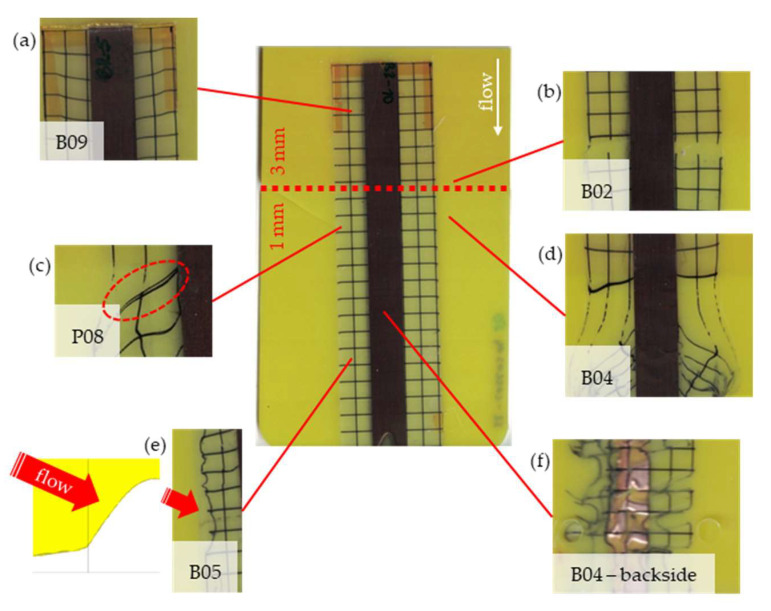
Overview of observed defects in regions (**a**–**f**). The red, dashed line indicates where the mold thickness changes from 3 to 1 mm.

**Figure 8 polymers-14-05060-f008:**
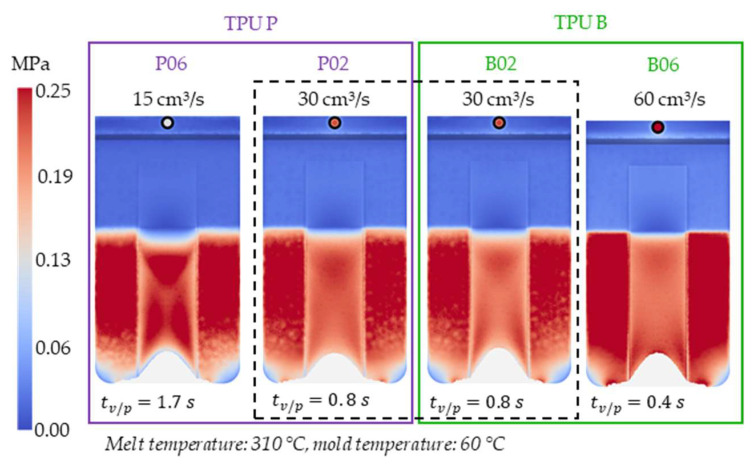
Shaded contour plots of the shear stress at wall result at switchover (t_v/p_ = 99% filled) for the settings with low melt and mold temperature.

**Figure 9 polymers-14-05060-f009:**
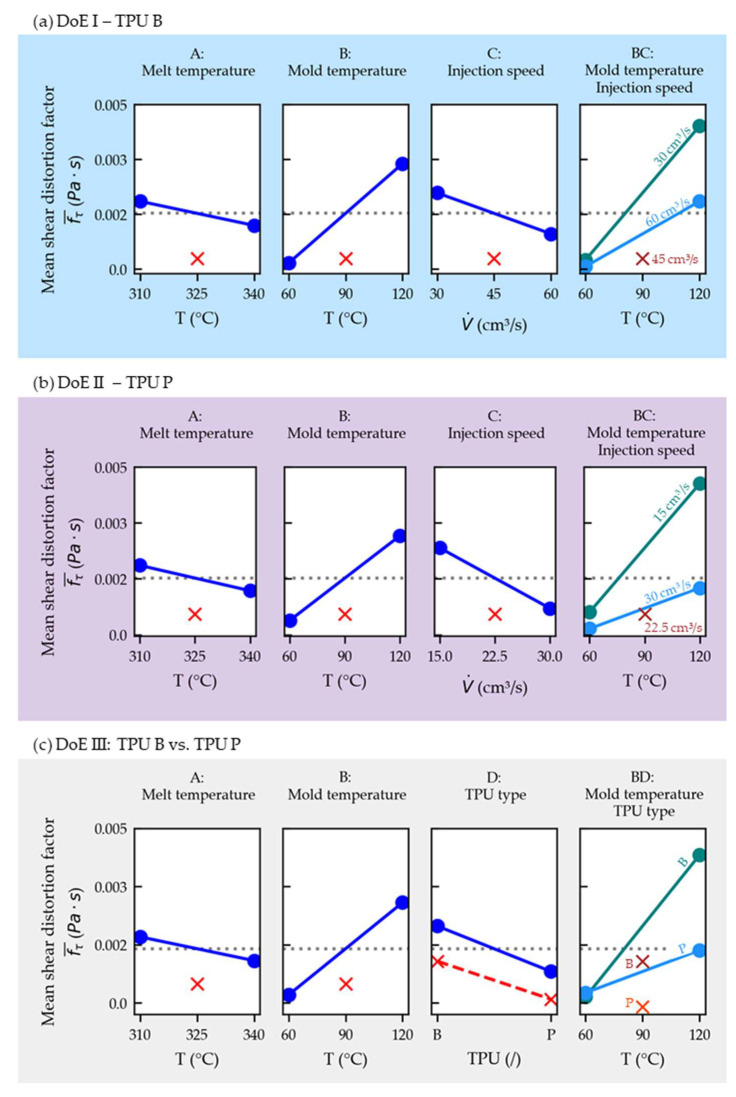
Main effects and interaction plots (including center points in red) for the averaged shear distortion factor ($\bar{f_{\tau }}$) for DoE I (**a**), II (**b**) and III (**c**). The factor designations (A, B, C and D) relate to Table 7.

**Figure 10 polymers-14-05060-f010:**
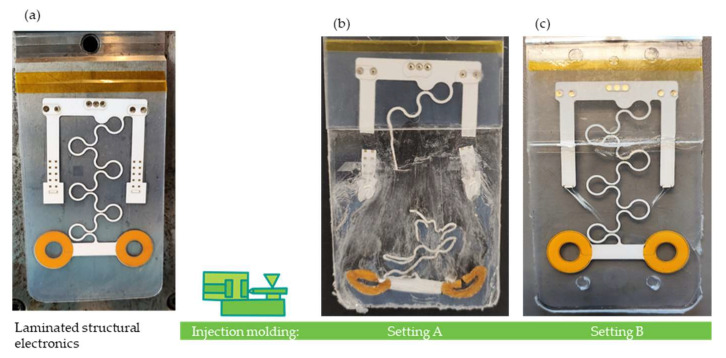
Smart@Surface demonstrator shown prior to overmolding (**a**) and after overmolding using two different sets of injection molding parameters (**b**,**c**). A summary of the used molding parameters is given in Table 8.

**Table 1 polymers-14-05060-t001:**
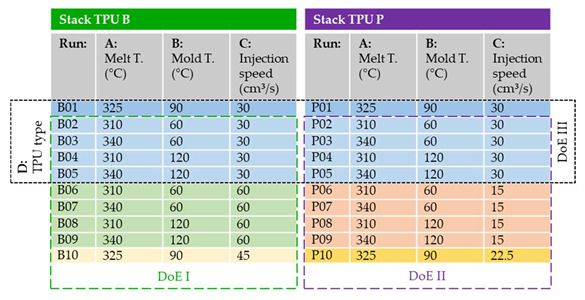
Experimental design for the overmolded parts with film strips. For each setting, 3 parts are produced (60 parts in total).

**Table 2 polymers-14-05060-t002:** Images of the overmolded films using TPU B for molding settings as outlined in Table 1. The associated distortion factor *f_d_* and damage category (I–III) are given below each image.

Stack TPU B				
B01 325 °C / 90 °C / 30 cm³/s 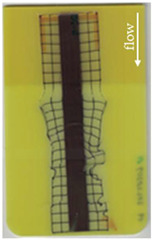 $f_{d}=0.85$ / III	B02 310 °C / 60 °C / 30 cm³/s 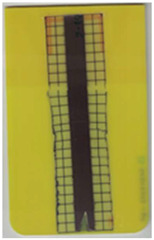 $f_{d}=0.27$ / II	B03 340 °C / 60 °C / 30 cm³/s 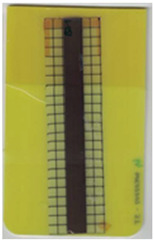 $f_{d}=0.06$ / I	B04 310 °C / 120 °C / 30 cm³/s  $f_{d}=1.00$ / III	B05 340 °C / 120 °C / 30 cm³/s 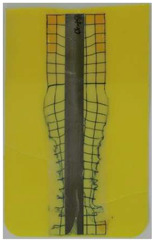 $f_{d}=0.89$ / III
B06 310 °C / 60 °C / 60 cm³/s  $f_{d}=0.04$ / I	B07 340 °C / 60 °C / 60 cm³/s 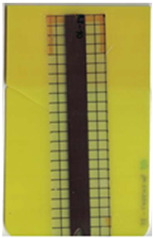 $f_{d}=0.02$ / I	B08 310 °C / 120 °C / 60 cm³/s  $f_{d}=0.93$ / III	B09 340 °C / 120 °C / 60 cm³/s  $f_{d}=0.43$ / II	B10 325 °C / 90 °C / 45 cm³/s 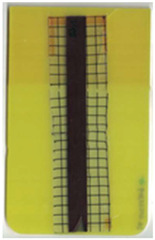 $f_{d}=0.29$ / II

**Table 3 polymers-14-05060-t003:** Images of the overmolded films using TPU P for molding settings as outlined in Table 1. The associated distortion factor *f_d_* and damage category (I–III) are given below each image.

Stack TPU P				
P01 325 °C / 90°C / 30 cm³/s 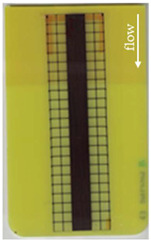 $f_{d}=0.02$ / I	P02 310 °C / 60 °C / 30 cm³/s 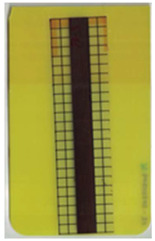 $f_{d}=0.00$ / I	P03 340 °C / 60 °C / 30 cm³/s  $f_{d}=0.00$ / I	P04 310 °C / 120 °C / 30 cm³/s 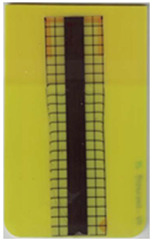 $f_{d}=0.27$ / II	P05 340 °C / 120 °C / 30 cm³/s  $f_{d}=$0.08 / I
P06 310 °C / 60 °C / 15 cm³/s 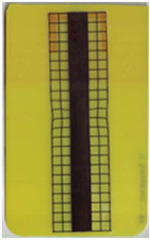 $f_{d}=$0.19 / II	P07 340 °C / 60 °C / 15 cm³/s 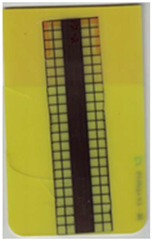 $f_{d}=0.04$ / I	P08 310 °C / 120 °C / 15 cm³/s 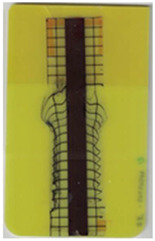 $f_{d}=0.69$ / III	P09 340 °C / 120 °C / 15 cm³/s  $f_{d}=0.25$ / II	P10 325 °C / 90 °C / 22.5 cm³/s 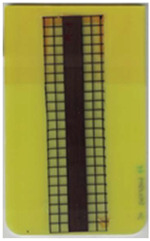 $f_{d}=0.07$ / I

**Table 4 polymers-14-05060-t004:** ANOVA of DoEs I, II and III in respect to the associated distortion factors *f_d_*.

Distortion Factors (*f_d_*):			
DoE I: Stack TPU B			
*p*-values for coefficients:	Melt temperature (A): 0.0% Mold temperature (B): 0.0% Injection speed (C): 0.0% Center point (CtPt): 1.5%		
Regression:	$f_{d}=1.9-7.0\;\times \;10^{-3}\;\cdot \;A+1.32\;\cdot \;B-6.6\;\times \;10^{-3}\;\cdot \;C-1.7\;\times \;10^{-1}\;\cdot \;CtPt$		
R^2^:	94%	R^2^_adj._:	93%
DoE II: Stack TPU P			
*p*-values for coefficients:	Melt temperature (A): 0.0% Mold temperature (B): 0.0% Injection speed (C): 0.0% Center point (CtPt): 5.5%		
Regression:	$f_{d}=2.3-6.6\;\times \;10^{-3}\;\cdot \;A+4.4\;\times \;10^{-3}\;\cdot \;B-1.4\;\times \;10^{-2}\;\cdot \;C-1.2\;\times \;10^{-1}\;\cdot \;CtPt$		
R^2^:	81%	R^2^_adj._:	78%
DoE III: Stack TPU B and TPU P at injection speed 30 cm^3^/s			
*p*-values for coefficients:	Melt temperature (A): 9.1% Mold temperature (B): 0.0% TPU type (D): 0.0% Center point (CtPt): 17.1%		
Regression:	$f_{d}=9.8-4.2\;\times \;10^{-3}\;\cdot \;A+8.0\;\times \;10^{-3}\;\cdot \;B-2.7\;\times \;10^{-1}\;\cdot \;D+1.1\;\times \;10^{-1}\;\cdot \;CtPt$		
R^2^:	83%	R^2^_adj._:	80%

**Table 5 polymers-14-05060-t005:** Shaded contour plots of the calculated shear distortion factor for film strips using TPU B for molding settings as outlined in Table 1.

Stack TPU B				
B01 325 °C / 90 °C / 30 cm³/s 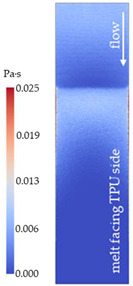 $\bar{f_{\tau }}=7.5\;\times \;10^{-4}\mathrm{Pa}\;\cdot \;s$	B02 310 °C / 60 °C / 30 cm³/s 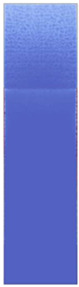 $\bar{f_{\tau }}=3.3\;\times \;10^{-4}\mathrm{Pa}\;\cdot \;s$	B03 340 °C / 60 °C / 30 cm³/s  $\bar{f_{\tau }}=2.4\;\times \;10^{-4}\mathrm{Pa}\;\cdot \;s$	B04 310 °C / 120 °C / 30 cm³/s 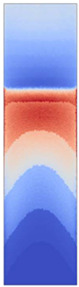 $\bar{f_{\tau }}=5.4\;\times \;10^{-3}\mathrm{Pa}\;\cdot \;s$	B05 340 °C / 120 °C / 30 cm³/s 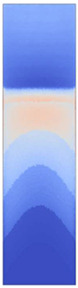 $\bar{f_{\tau }}=3.3\;\times \;10^{-3}\mathrm{Pa}\;\cdot \;s$
B06 310 °C / 60 °C / 60 cm³/s  $\bar{f_{\tau }}=9.3\;\times \;10^{-5}\mathrm{Pa}\;\cdot \;s$	B07 340 °C / 60 °C / 60 cm³/s  $\bar{f_{\tau }}=6.9\;\times \;10^{-5}\mathrm{Pa}\;\cdot \;s$	B08 310 °C / 120 °C / 60 cm³/s 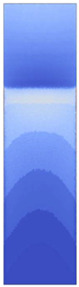 $\bar{f_{\tau }}=2.5\;\times \;10^{-3}\mathrm{Pa}\;\cdot \;s$	B09 340 °C / 120 °C / 60 cm³/s 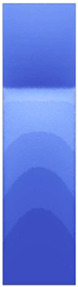 $\bar{f_{\tau }}=1.7\;\times \;10^{-3}\mathrm{Pa}\;\cdot \;s$	B10 325 °C / 90 °C / 45 cm³/s 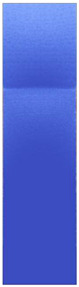 $\bar{f_{\tau }}=3.2\;\times \;10^{-4}\mathrm{Pa}\;\cdot \;s$

**Table 6 polymers-14-05060-t006:** Shaded contour plots of the calculated shear distortion factor for film strips using TPU P for molding settings as outlined in Table 1.

Stack TPU P				
P01 325 °C / 90 °C / 30 cm³/s 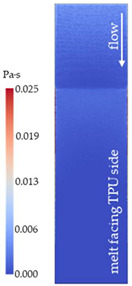 $\bar{f_{\tau }}=3.4\;\times \;10^{-4}\mathrm{Pa}\;\cdot \;s$	P02 310 °C / 60 °C / 30 cm³/s  $\bar{f_{\tau }}=2.1\;\times \;10^{-4}\mathrm{Pa}\;\cdot \;s$	P03 340 °C / 60 °C / 30 cm³/s 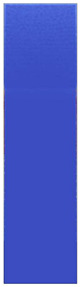 $\bar{f_{\tau }}=1.5\;\times \;10^{-4}\mathrm{Pa}\;\cdot \;s$	P04 310 °C / 120 °C / 30 cm³/s 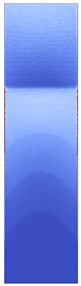 $\bar{f_{\tau }}=1.7\;\times \;10^{-3}\mathrm{Pa}\;\cdot \;s$	P05 340 °C / 120 °C / 30 cm³/s 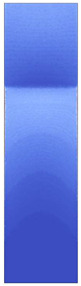 $\bar{f_{\tau }}=1.1\;\times \;10^{-3}\mathrm{Pa}\;\cdot \;s$
P06 310 °C / 60 °C / 15 cm³/s 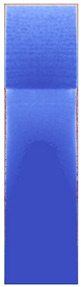 $\bar{f_{\tau }}=7.8\;\times \;10^{-4}\mathrm{Pa}\;\cdot \;s$	P07 340 °C / 60 °C / 15 cm³/s 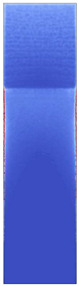 $\bar{f_{\tau }}=5.7\;\times \;10^{-4}\mathrm{Pa}\;\cdot \;s$	P08 310 °C / 120 °C / 15 cm³/s 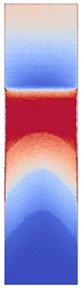 $\bar{f_{\tau }}=5.6\;\times \;10^{-3}\mathrm{Pa}\;\cdot \;s$	P09 340 °C / 120 °C / 15 cm³/s 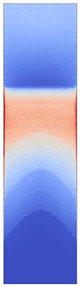 $\bar{f_{\tau }}=3.4\;\times \;10^{-3}\mathrm{Pa}\;\cdot \;s$	P10 325 °C / 90 °C / 22.5 cm³/s 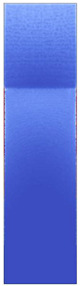 $\bar{f_{\tau }}=6.2\;\times \;10^{-4}\mathrm{Pa}\;\cdot \;s$

**Table 7 polymers-14-05060-t007:** Regression values of DoEs I, II and III for the shear distortion factors ($\bar{f_{\tau }}$).

Averaged Shear Distortion Factor ($\bar{f_{\tau }}$):			
DoE I: Stack TPU B			
Factors:	Melt temperature (A) Mold temperature (B) Injection speed (C) Center point (CtPt)		
Regression:	$\bar{f_{\tau }}=2.3\;\times \;10^{-3}-2.4\;\times \;10^{-5}\;\cdot \;A+1.0\;\times \;10^{-4}\;\cdot \;B+6.3\;\times \;10^{-5}\;\cdot \;C-1.0\;\times \;10^{-6}\;\cdot \;BC-1.4\;\times \;10^{-6}\;\cdot \;CtPt$		
R^2^:	95.33%	R^2^_adj._:	87.55%
DoE II: Stack TPU P			
Factors:	Melt temperature (A) Mold temperature (B) Injection speed (C) Center point (CtPt)		
Regression:	$\bar{f_{\tau }}=2.9\;\times \;10^{-3}-2.5\;\times \;10^{-5}\;\cdot \;A+1.1\;\times \;10^{-4}\;\cdot \;B+1.4\;\times \;10^{-4}\;\cdot \;C-3.0\;\times \;10^{-6}\;\cdot \;BC-1.1\;\times \;10^{-3}\;\cdot \;CtPt$		
R^2^:	94.48%	R^2^_adj._:	85.28%
DoE III: Stack TPU B and TPU P at injection speed 30 cm^3^/s			
Factors:	Melt temperature (A) Mold temperature (B) TPU type (D) Center point (CtPt)		
Regression:	$\bar{f_{\tau }}=5.0\;\times \;10^{-3}-2.3\;\times \;10^{-5}\;\cdot \;A+4.4\;\times \;10^{-5}\;\cdot \;B+1.5\;\times \;10^{-3}\;\cdot \;D-2.4\;\times \;10^{-5}\;\cdot \;BD-1.0\;\times \;10^{-3}\;\cdot \;CtPt$		
R^2^:	93.35%	R^2^_adj._:	85.03%

**Table 8 polymers-14-05060-t008:** Summary table with molding parameters used to manufacture the Smart@Surface demonstrators shown in Figure 10b,c.

	Melt Temperature (°C)	Mold Temperature (°C)	Injection Speed (cm^3^/s)	TPU Type (/)
Setting A	310	120	30	B
Setting B	340	120	60	P

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
