# Peer review of "Process Optimization of Injection Overmolding Structural Electronics with Regard to Film Distortion"

_polymers, 2022, doi:10.3390/polym14235060_

Round 1
Reviewer 1 Report
In general, the article is scientific. It investigated the effects of injection process parameters on the quality of overmolding film. And this article explained the effects by experiments and simulations. Although there are some details should be modified, the main part is logical and substantial. Overall, I would suggest the publication of this manuscript.
Minor comments and suggestions that the authors may want to address.
Comment 1: In lines 145 and 146, the thicknesses of the TPU B and the TPU P in the second stack was different. Please explain the reason for this design.
Comment 2: In lines 190-192, it was said that the appropriate injection speeds of TPU B films and TPU P films, however, it was not clear how to evaluate the quality of these films. Things like how much damage, warpage or other defects were not acceptable.
Comment 3: In line 199, it was said that B02-B10 and P02-P10 was the process parameters of the films, but I think it should be B02-B09 and P02-P09, please check.
Comment 4: The injection speed below 60 cm3/s showed damage parts for TPU B, but the low-level injection speed was 30 cm3/s in the experiments, please explain the reason for choosing this injection speed. So does the injection speed of TPU P parts.
Comment 5: In the end of section 3.1, it would better to show the compare results by pictures of TPU B parts and TPU P parts.
Comment 6: There were many “section 0”in the article, and they did not mean the same things. Please explain the meaning of section 0 in each place.
Comment 7: In section 3.3, it was better to explain the reason why the injection process parameters were set as table 8.
Author Response
Dear reviewer!
Thank you very much for your very precise and specific comments! I’m trying to address them one by one in the following:
Comment 1: It was – unfortunately – not possible to acquire both TPUs sheets in the same thickness. We tried to laminate a 25 µm Platilon TPU on top of a 75 µm Platilon TPU to achieve the same thickness of 100 µm as with the Bemis TPU. However this resulted in major wrinkles between the tow TPU layers during lamination and we had to abandon this approach. Different stack thicknesses will have an impact upon overmolding. Hence we addressed this e.g. in Figure 8 (line 383) where we show that the shear stresses are still comparable in their magnitude.
Comment 2: Indeed we do not go into details about how we have evaluated the parts produced during the preliminary tests (line 190-192). The aim of the preliminary test was solely to explore the processing window and hence to find possible DoE factor levels (between which differences could be observed). It became clear that we could not set up a meaningful DoE comprising both TPU types at two injection speeds. (A film using TPU B overmolded at 15 ccm/s will result in a completely destroyed part with an f_d close to 1 regardless of the other processing parameters). Right below, in line 193ff we explain the set up for the used DoEs. We then introduce the shear distortion factor f_d for the parts evaluated in the manuscript (line 217-222).
Comment 3: DoE I contains settings B02-B10 (8+1 settings). DoE II contains settings P02-P10 (8+1 settings). DoE III contains settings B01-B05 and P01-P05 (8+2 settings). This is illustrated in Table 1 (line 211). B10 is the center point for DoE I and P10 is the center point for DoE II. B01 and P01 are the center points for DoE III. We double checked the text and it should be fine.
Comment 4: The factor levels for the DoEs were deliberately chosen in a way so that good (low f_d) and bad (high f_d) parts could be produced on purpose. (This was also the reason for the preliminary test addressed in Comment 2.) Regarding the injection speeds for parts with TPU B the levels were hence set to 30 and 60 ccm/s to have the full range of good and bad parts (and to 15 and 30 ccm/s for parts with TPU P to achieve the same). If we had only used the injection speed of e.g. 60 ccm/s for parts with TPU B we would not have been able to assess the magnitude of the effect of the injection speed in relation to the other factors (cf. Figure 6 a and b). If we only had used 30 and 60 ccm/s for the TPU P, we would just have obtained almost exclusively good parts and hardly any distinction between them would have been possible.
Comment 5: Table 2 shows the parts produced using TPU B and Table 3 those with TPU P. This was done so that the influence of the processing settings can be observed easily. Parts that are part of DoE III, comparing the two TPUs as factor, are depicted with a blue background to aid comparison. We therefore do not agree that it would be better to show the compare results by pictures of TPU B parts and TPU P parts.
Comment 6: No “section 0” were present in the version that we uploaded. We assume that they have been formed during the upload by the journal automatically. We have changed all the “section 0” to their original and intended Cross-reference.
Comment 7: We feel that we have explained the reasons for choosing the processing parameters in line 441-443. We used the processing settings according to the relations found and described in the manuscript. However a high mold temperature was used for a smoother depiction of the mold surface. To make the decision of the process settings even more transparent, we added the information that setting A corresponds to the settings of the DoE, which produced bad parts and setting B corresponds to the settings of the DoE, which produced good parts in order to prove that these settings also produce similar results in the real demonstrator part compared to the simple test part, on which the DoE was performed.
With best regards.
Reviewer 2 Report
This manuscript reported Process optimization of injection overmolding structural electronics with regards to film distortion. The effects of the molding settings on the flexible laminated electronics were investigated. However, this paper needs to undergo minor revisions before publication.
Some major concerns are shown as following:
1. Introduction: In this section, a fuller review of the relevant work is needed.
2. The authors should include the novelty of the manuscript.
3. The mechanism analysis of the process optimization of injection overmolding structural electronics should be added.
4. Please reformat all tables according to the latest articles in Polymers.
5. Please unify the font and format of the unit in the text.
6. There must be a space between the number and the unit.
7. The format of the references is not uniform, please refer to the latest articles in Polymers to modify the format of the references.
8. The authors need to cite some articles published in Polymers.
Author Response
Dear reviewer!
Thank you for your comments! I’m trying to address them one by one in the following:
Comment 1: We double checked for missing literature but could not find articles that we feel should be included. If you could point us to specific references that should be included, then we would be happy to consider those.
Comment 2: We feel that the novelty of the manuscript was stressed sufficiently in line 121-125.
Comment 3: We perhaps do not fully understand your comment. The optimization was based on DoEs, which were evaluated using experiments (factor f_d) and simulations (factor f_tau). We concluded (that in respect to film distortion) the overmolding of structural electronics could be optimized by aiming for settings that yield low f_d and f_tau.
Comment 4: We contacted MDPI about the formatting of our tables prior to submission. They encouraged us to go ahead with our design and wrote us that the editors would help us with possible required design adaptations.
Comment 5: We changed the word-formulas within the main text to text using font “Palatino Linotype”.
Comment 6: We double checked. There (now) should be a space between each number and unit. However for the percentage sign, we did not use the ISO 31-0 style and went with the common practice of not using a space between the number and the percentage sign (which we also found in all of the about 5 arbitrarily chosen recent MDPI Polymers articles.)
Comment 7: We have used Citavi’s Polymers (MDPI) template. We expect the editors to address possible citation style issues prior to publication.
Comment 8: This comment is in contradiction to the reviewer guidelines, published at the MDPI webpage (section 7.4 Review Reports https://www.mdpi.com/reviewers#_bookmark11):
“Reviewers must not recommend citation of work by themselves, close colleagues, another author, or the journal when it is not clearly necessary to improve the quality of the manuscript under review.”
and
“Reviewers must not recommend excessive citation of their work (self-citations), another author’s work (honorary citations) or articles from the journal where the manuscript was submitted as a means of increasing the citations of the reviewer/authors/journal. You can provide references as needed, but they must clearly improve the quality of the manuscript under review”.
However, if there were relevant and missing articles that happened to have been published in Polymers, then we would be happy to include those. (See also comment 1.)
With best regards.